# Improving the Climate Resilience of Rice Farming in Flood-Prone Areas through Azolla Biofertilizer and Saline-Tolerant Varieties

Tualar Simarmata [1],*, Muhamad Khais Prayoga [2],*, Mieke R. Setiawati [1], Kustiwa Adinata [3] and Silke Stöber [4]

1 Department of Soil Sciences and Land Resources, Faculty of Agriculture, Universitas Padjadjaran, Bandung 45363, Indonesia; m.setiawati@unpad.ac.id
2 Research Institute for Tea and Cinchona, Bandung 45363, Indonesia
3 Indonesian Farmers Community Network (JAMTANI), Pangandaran 46396, Indonesia; kustiwa.adinata@gmail.com
4 Centre for Rural Development (SLE), Humboldt-Universität zu Berlin, 10117 Berlin, Germany; silke.stoeber@agrar.hu-berlin.de
* Correspondence: tualar.simarmata@unpad.ac.id (T.S.); mkprayoga@iritc.org (M.K.P.)

**Abstract:** Rice farming in coastal areas is often victim to flooding as a result of climate change. Low-cost adaptation strategies are required to increase resilience and rice productivity in these flood-prone coastal areas. In this study, enriched Azolla extract (EAE) liquid biofertilizers, combined with selected stress-tolerant rice varieties, were tested in farmers' fields in Pangandaran, West Java from June to October 2020. This study aimed to investigate the effectiveness of EAE in increasing the yield of different rice varieties. The research was arranged as a split-plot design with five replications. The main plot was the EAE application (T1 = 3 ton ha$^{-1}$ compost and T2 = 3 ton ha$^{-1}$ compost + 10 L ha$^{-1}$ of EAE), and the sub-plots were stress-tolerant rice varieties (V1 = Inpari 43, V2 = Mawar, V3 = Inpari 30, V4 = Inpara 03, V5 = Mendawak). The application of EAE of 10 L ha$^{-1}$ significantly affected the rice grain yield, which was 37.06% higher than that of the control plot. The average grain yield of the five varieties under EAE treatment (5.51 ton ha$^{-1}$) was greater than the grain yield of local farmers' fields (3.78−4.97 ton ha$^{-1}$). Inpari 43 had the highest grain yield with 5.90 ton ha$^{-1}$, but the yield was not significantly different from the Mendawak variety (4.90 ton ha$^{-1}$). This result suggests that EAE and selected stress-tolerant rice varieties (Inpari 43 or Mendawak) are an effective adaptation strategy to increase rice farms' resilience and productivity in coastal areas prone to flooding.

**Keywords:** enriched Azolla extract (EAE); liquid biofertilizer; saline-tolerant varieties; rice farming; flood; coastal area; climate change; Indonesia

## 1. Introduction

The increase in rice production in Indonesia is constrained by various factors, one of which is climate change. There is no longer a debate about whether or not climate change exists, but rather about how countries, institutions, and communities can best adapt to the severe impacts of the climate crisis [1]. In the rice farming context, coastal regions are the areas most affected by climate change.

One impact of climate change is the change in thermal temperature in the Indian Ocean which increases the intensity of rainfall on the southern coast of Java. The increase in rainfall intensity in the southern coastal areas of Java such as Pacitan, Trenggalek, and Tulung Agung reached 18−23 mm per year [2]. Similarly, the results of data analysis sourced from the Indonesian meteorological agency (2018) showed that rainfall around the Cilacap weather station (generalizable to both regions, Cilacap and Pangandaran) increased by 22 mm per year [3]. Increased rainfall increases the risk of flooding in the rice fields.

The resultant flooding of the rice fields causes more frequent and longer periods of inundation of the rice plants, in turn causing abiotic stress which adversely affects the growth of the rice plants [4]. The effects of inundation are complex and varied. One

of the effects of inundation is the inhibition of rice plants' metabolism, characterized by leaf chlorosis [5]. In flooded conditions, rice plant respiration will turn into anaerobic respiration due to the stress caused by lack of oxygen (hypoxia). Hypoxia can trigger plant cellular responses that result in a decrease in plant cellular pH, which can cause plant cell death [6]. Moreover, the regular inundation of rice fields causes a decrease in soil fertility due to nutrient leaching [7].

Flooding is also a limiting factor for effective fertilizer application. Fertilizer is usually immersed into the soil; however, this is difficult under flooded conditions [5]. Therefore, to overcome this problem, liquid fertilizers are applied to flooded rice fields and sprayed on plants' stems and leaves, which is commonly known as foliar feeding. Fertilization by foliar feeding should ideally involve a liquid fertilizer that is highly effective and is not harmful to the environment. Therefore, the use of liquid biofertilizer from organic matter is regarded as one important mitigation strategy for combating climate change [7,8].

The use of organic fertilizers is a prime example of an agricultural practice which supports sustainable agroecosystems and improves soil quality [9]. Green manure is one such organic material which may be used as a liquid fertilizer, as it is easier to process into liquid form than manure [10]. In Indonesia, there are various green manure plants which can be processed into liquid fertilizer, such as Gliricidia sepium, Chromolaena odorata, Leucaena leucocephala, Sesbania rostrata, and Azolla pinnata. Each green manure has a different nutrient profile, as presented in Table 1. Azolla pinnata has the highest nitrogen content of 5.30%, while Sesbania rostrata contains the most potassium at 4.56%.

Green manure plants with a low lignin and cellulose content are considered the best raw materials for liquid fertilizers. However, most plant biomass material is lignocellulosic, and almost half of the lignocellulosic material is cellulose and lignin compounds. The degradation of lignin is a limiting step for the speed and efficiency of decomposition because lignin acts as a barrier to the penetration of the enzyme solution. Its complex structure, high molecular weight, and insoluble nature in water make lignin degradation very difficult; as such, lignin is considered a barrier to cellulolytic enzyme access [11]. Azolla, is in this context, is more favorable, as it contains 1.5% lignin and 14.08% cellulose, while Sesbania plants contain almost twice that amount (2.5% lignin and 27.30% cellulose) [12].

**Table 1.** Nitrogen, phosphorus, and potassium content in various green manure plants.

| Green Manure | Nitrogen (%) | Phosphorus (%) | Potassium (%) |
|---|---|---|---|
| Gliricidia sepium | 3.09 | 0.18 | 0.23 |
| Chromolaena odorata | 2.48 | 0.34 | 0.61 |
| Leucaena leucocephala | 3.01 | 0.21 | 0.30 |
| Sesbania rostrata | 3.70 | 0.24 | 4.56 |
| Azolla pinnata | 5.30 | 0.87 | 3.14 |

Source: [10,12].

In general, Azolla pinnata is relatively more suitable as a liquid fertilizer than Gliricidia sepium, Chromolaena odorata, Leucaena leucocephala, and Sesbania rostrata. Azolla pinnata, which lives in symbiosis with Anabaena azollae, is able to fix nitrogen (N) from the air. In this regard, it can be used as a substitute for urea fertilizer (the most common inorganic fertilizer used by farmers), and therefore can be considered a climate change mitigation method. The application of Azolla pinnata biomass can lessen the use of urea fertilizer by 50–80 kg. By applying Azolla biomass, as much as 10 tons ha$^{-1}$, the organic carbon (C-organics) content of the soil can be increased by 0.30%, and the nitrogen content by 0.05%. In addition, the use of 10 tons ha$^{-1}$ of Azolla can increase grain yield by 6.19% [13].

Increasing rice productivity in coastal areas that are stressed by inundation due to climate change requires further efforts beyond just fertilization through foliar feeding. Foliar feeding needs to be combined with the use of rice varieties that are tolerant to inundation and salinity stress. Inundation caused by tidal flooding due to rising sea levels

brings in salt particles, increasing soil salinity. High levels of soil salinity can inhibit the growth and development of rice plants, which requires the use of salinity tolerant varieties in coastal areas [3].

The Ministry of Agriculture of the Republic of Indonesia has released several varieties of rice which are tolerant to inundation stress and salinity. However, these varieties have yet to be introduced into communities to test their adaptation potential for more resilient rice production in coastal regions. The purpose of this study was to test and obtain the best combination of enriched Azolla extract (EAE) and saline-tolerant rice varieties to increase rice farming resilience in a coastal area where climate change has already intensified flooding and its associated risks to rice farming.

## 2. Materials and Methods

The study was conducted in Paledah Village, Padaherang District, Pangandaran Regency, and West Java Province, Indonesia from June to October 2020. The research was designed as a split-plot with five replications. The main plot consisted of the enriched Azolla extract (EAE) application with subplots of five stress-tolerant rice varieties (Table 2). This study followed the principles of participatory farmer-led research and involved five farmers from the farmer organization JAMTANI.

**Table 2.** Split-plot design.

| Plots | Treatments | Description |
|---|---|---|
| Main plots | T1 | 3 ton ha$^{-1}$ compost (positive control) |
|  | T2 | 3 ton ha$^{-1}$ compost + 10 L ha$^{-1}$ EAE |
| Subplots | V1 | Inpari 43 |
|  | V2 | Mawar |
|  | V3 | Inpari 30 |
|  | V4 | Inpara 03 |
|  | V5 | Mendawak |

The participatory farmer-led research method applies technologies within the farming environment and involves farmers in the entire field research process. This encourages the farmers' adoption of the new technologies [14]. This farmer-led research involved farmers not only in the design, but also in data collection and analysis. The study was implemented through a farmer field school approach. Farmers met with the scientists and field school facilitators every two weeks to record their observations and present them to their fellow farmers. The participating farmers made their observations according to a standardized protocol.

The parameters observed included plant height (cm), number of productive tillers, panicle length (cm), number of pithy grains per panicle, weight of 100 grains (grams), grain weight per plant (grams), and grain yield (ton ha$^{-1}$). Differences between the varieties were tested via an ANOVA post hoc analysis using Tukey's honest significance test (Tukey's HSD). The data were computed using the PKBT-STAT 4.0 software. For comparison, the research team compared grain yields of the split-plot with yields obtained by local farmers in the surrounding field who farm conventionally by using inorganic fertilizers.

## 3. Results and Discussion

The results of the analysis of variance (ANOVA) for each parameter indicate that there was no interaction between EAE treatments and varieties. From this, we can conclude that all five observed parameters are also influenced by independent factors; both treatment and rice variety. If no interaction between the treatment and the variety can be recorded, another test must be performed, namely the average treatment test or the average variety test [15]. The results of the average treatment test reveal that almost all parameters differed significantly from each other ($p \leq 0.05$) except for the plant height characteristic. The

results of the average variety test are even clearer: all characteristics show very significant differences ($p \leq 0.01$) (Table 3).

**Table 3.** Analysis of variance for each parameter.

| Parameters | Value of $F_{count}$ | | | CV (%) |
|---|---|---|---|---|
| | Treatments x Varieties | Treatments | Varieties | |
| Plant height | 0.10 (NS) | 3.04 (NS) | 4.57 (**) | 8.73 |
| Number of productive tillers | 0.45 (NS) | 7.51 (*) | 5.96 (**) | 14.43 |
| Weight of 100 grains | 1.61 (NS) | 7.22 (*) | 35.37 (**) | 7.97 |
| Grain weight per plant | 0.60 (NS) | 6.02 (*) | 4.93 (**) | 15.27 |
| Grain yield | 0.60 (NS) | 6.02 (*) | 4.93 (**) | 15.27 |

\* Significant in $p \leq 0.05$; \*\* significant in $p \leq 0.01$; NS = non-significant; CV = coefficient of variation.

Moreover, Table 3 reveals that the coefficient of variation (CV) ranges from 7.97 to 15.27%. The value of the CV shows the level of accuracy of the compared treatments. A CV value of below 20% indicates that the experimental error for the observed characteristics is relatively small [16]. Based on this, the CV value in this study is classified as relatively good with a small experimental error.

The parameter with the second lowest CV value of 8.73% is plant height. Plant height is a good indicator of growth, showing the influence of the environment or the treatment applied [17]. Based on the ANOVA results, there was no significant difference in the enriched Azolla extract (EAE) treatment, but there was a very significant difference in the varieties. This shows that EAE does not have a significant effect on the characteristic of plant height (Table 4), so the differences in plant height are mainly caused by genetic differences of each variety.

**Table 4.** Comparison of growth and production parameters of the treatments.

| Parameters | Treatments | |
|---|---|---|
| | T1 | T2 |
| Plant height (cm) | 98.08 [A] | 103.20 [A] |
| Number of productive tillers | 18.52 [B] | 26.12 [A] |
| Weight of 100 grains (gram) | 3.26 [B] | 3.45 [A] |
| Grain weight per plant (gram) | 57.40 [B] | 78.66 [A] |
| Grain yield (ton ha$^{-1}$) | 4.02 [B] | 5.51 [A] |

Note: Numbers followed by the same letter are non-significant ($p \leq 0.05$); T1 = 3 tons ha$^{-1}$ compost; T2 = 3 tons ha$^{-1}$ compost + 10 L ha$^{-1}$ EAE.

Plant height, apart from being an indicator of growth, shows adaptation to inundation. Elongation during inundation is an avoidance strategy which works by initiating aerobic metabolism to keep the shoots above the water surface [18,19]. The results showed that the Mawar variety had a fairly high average plant height (108.60 cm) but was not significantly different from the Inpari 43, Inpara 03, and Mendawak varieties based on the results of a further Tukey's HSD test (Table 5).

**Table 5.** Comparison of growth and production parameters of the tested rice varieties.

| Parameters | Varieties | | | | |
|---|---|---|---|---|---|
| | Inpari 43 | Mawar | Inpari 30 | Inpara 03 | Mendawak |
| Plant height (cm) | 102.60 [ab] | 108.60 [a] | 92.10 [b] | 100.00 [ab] | 99.90 [ab] |
| Number of productive tillers | 32.30 [a] | 19.80 [b] | 18.50 [b] | 17.90 [b] | 23.10 [ab] |
| Weight of 100 grains (gram) | 4.09 [a] | 3.33 [b] | 2.70 [c] | 3.19 [b] | 3.46 [b] |
| Grain weight per plant (gram) | 98.33 [a] | 56.53 [b] | 44.57 [b] | 59.01 [ab] | 81.72 [a] |
| Grain yield (ton ha $^{-1}$) | 6.88 [a] | 3.96 [b] | 3.12 [b] | 4.13 [ab] | 5.72 [a] |

Note: Numbers followed by the same letter are non-significant ($p \leq 0.05$).

The average plant height of all varieties ranged from 92.10 to 108.60 cm. Farmers from the Pangandaran area prefer rice varieties with medium plant height (90−115 cm); this is because these varieties can readily adapt to both submersion and strong gusts of wind. Short plants (<90 cm) will be submerged during floods, while tall plants (>115 cm) will easily collapse under the force of the wind [20]. In general, the five varieties tested were in accordance with the preferences of farmers in the area based on the characteristic of plant height. In addition to plant height, this study also observed the parameters of yield and yield components. One of the observed yield component characteristics is the number of productive tillers. Only some rice tillers become productive by entering the generative phase and producing panicles [21]. Based on the ANOVA results, the number of productive tillers showed a significant difference ($p \leq 0.05$) in the EAE treatment. Besides that, the characteristic showed a very significant difference ($p \leq 0.01$) in the variety parameter (Table 3).

Based on the results of the Tukey's HSD follow-up test, the average number of productive tillers in T2 (EAE treatment) was higher than in T1 (positive control of 3 tons ha$^{-1}$ compost without EAE treatment). A larger number of productive tillers leads to higher rice productivity. The nutrient supply in the soil plays an important role in the process of forming productive tillers and is maximized under optimal soil fertility conditions [22]. Thus, it is presumed that EAE increases the nutrient content in the soil, which can then be absorbed by the rice plant and used to produce productive tillers. Table 3 shows that the variety Inpari 43 (32.30) had the highest average number of productive tillers compared to Mawar (19.80), Inpari 30 (18.50), and Inpara 03 (17.90) varieties. The Mendawak variety had on average 23.10 productive tillers and did not differ significantly from Inpari 43.

Another important yield component characteristic is the weight of 100 grains, which indicates the ability of rice plants to maximize the utilization of nutrients as food reserves. The process of grain filling greatly determines the potential yield of rice plants. The number of productive tillers is therefore important but does not automatically produce a high yield if the grains remain small or empty [3]. Therefore, the weight of 100 grains parameter is important; it determines the level of effectiveness of nutrient absorption. The ANOVA results showed that there was a significant difference ($p \leq 0.05$) in the EAE treatment, and a very significant difference ($p \leq 0.01$) between the varieties. Based on Tukey's HSD further test, the 100 grain weight under EAE treatment T2 weighed 0.19 grams more than the positive control T1 treatment (Table 4). Thus, the addition of 10 L ha$^{-1}$ of EAE was able to increase the weight of 100 grains by 5.85%.

In general, compared with the variety descriptions of the five rice varieties studied, the 100 grain weight obtained in the experimental plot was higher. The variety description issued by the Ministry of Agriculture of the Republic of Indonesia indicates for Inpari 43 2.73 grams per 100 grains, Mawar, 2.62 grams, Inpari 30, 2.41 grams, Inpara 03, 2.68 grams, and Mendawak, 2.84 grams. In this study, Inpari 43 achieved 4.09 grams, the highest 100 grain weight of all the tested varieties (Table 5). This means that the 100 grain weight of Inpari 43 was 49.82% higher than the variety description. Accordingly, it is suspected that the Inpari 43 variety can adapt effectively to an environment that is stressed by inundation and salinity. If the cultivation is additionally supported with EAE soil fertility treatment, a high yield can be achieved, even under adverse environmental conditions.



Both of these parameters, the number of productive tillers and the 100 grain weight, affect the grain weight per plant. With more productive tillers and larger grains, the grain weight per plant increases [23]. As stated earlier, panicle formation is strongly influenced by nutrient availability, of which nitrogen is crucial [24]. The ANOVA results reveal that the grain weight per plant in the EAE treatment (T2) was higher (78.66 grams) than that of the control (T1) with 57.40 grams (Table 4). By applying 10L ha$^{-1}$ EAE, the grain weight per plant increased by 37.04%.

Grain weight per plant positively correlated with the parameter grain yield. The higher the weight per plant, the higher the grain yield [3]. The ANOVA results disclose that there was no interaction between the treatment and varieties, but there was a significant difference ($p \leq 0.05$) in the EAE treatment and a very significant difference ($p \leq 0.01$) in the variety. Therefore, it is presumed that the differences are caused by the independent influence of the treatment and variety. Based on Tukey's HSD test, the EAE treatment (T2) had an average grain yield of 5.51 tons ha$^{-1}$, which was higher than the control (T1) with an average grain yield of 4.02 ton ha$^{-1}$ (Table 4).

To investigate the effectiveness of enriched Azolla extract (EAE), production was compared with harvest data from farmers producing rice under similar environmental conditions near the study site. Data were collected from five farmers. All farmers use synthetic fertilizers, with two farmers (farmer 4 and 5) combining it with organic fertilizers (Table 6). The grain yields from local farmers ranged from 3.78−4.97 tons ha$^{-1}$. Farmer 5, with highest grain yields of 4.97 ton ha$^{-1}$, planted the variety IR 64 with a mixed soil fertility strategy which included inorganic fertilizers and compost. The lowest yield was 3.78 ton ha$^{-1}$, where farmer 2 planted the Ciherang variety and only used inorganic fertilizers (Table 6).

**Table 6.** Local farmers' grain yield.

| Location | Varieties | Fertilizer | | Grain Yield (ton ha$^{-1}$) |
|----------|-----------|------------|---|---------------------------|
| Farmer 1 | IR 64 | 1. | Urea 200 kg ha$^{-1}$ | 4.34 |
|          |          | 2. | SP36 70 kg ha$^{-1}$ | |
|          |          | 3. | KCL 70 kg ha$^{-1}$ | |
| Farmer 2 | Ciherang | 1. | Urea 150 kg ha−1 | 3.78 |
|          |          | 2. | SP36 50 kg ha−1 | |
|          |          | 3. | KCL 50 kg ha−1 | |
| Farmer 3 | Mawar | 1. | Urea 300 kg ha$^{-1}$ | 4.20 |
|          |          | 2. | SP36 100 kg ha$^{-1}$ | |
|          |          | 3. | KCL 100 kg ha$^{-1}$ | |
| Farmer 4 | Ciherang | 1. | Urea 200 kg ha$^{-1}$ | 4.76 |
|          |          | 2. | SP36 75 kg ha$^{-1}$ | |
|          |          | 3. | KCL 75 kg ha$^{-1}$ | |
|          |          | 4. | Compost 1 ton ha$^{-1}$ | |
| Farmer 5 | IR 64 | 1. | Urea 250 kg ha$^{-1}$ | 4.97 |
|          |          | 2. | SP36 80 kg ha$^{-1}$ | |
|          |          | 3. | KCL 80 kg ha$^{-1}$ | |
|          |          | 4. | Compost 3 ton ha$^{-1}$ | |

In general, when compared to the yields from the five farmers around the study site, treatment T2 (EAE 10 L ha1) resulted in higher grain yield (Figure 1). Treatment T2 had a grain yield 26.96% higher than farmer 1, 45.77% higher than farmer 2, 31.19% higher than

farmer 3, 15.76% higher than farmer 4, and 10.87% higher than farmer 5. The comparison between the treatments in the study (T1 and T2) and the five surrounding farmers was not done statistically, so the level of difference in the treatments could not be quantified. This is one of the weaknesses of this study. Therefore, in future research, it would be better if comparisons with local farmers could be included in the research design so that the level of difference between treatments and nearby farmers could be quantified.

This study shows that EAE is very influential on the growth and production of rice plants. This is presumably due to the effect of EAE on increasing soil fertility which supports plant growth and development. Azolla is able to increase the nitrogen and C-organic content in the soil, both of which are relevant soil fertility indicators [17]. The application of long-term organic matter greatly affects the enzymatic activity of the soil and is therefore positively correlated with the level of soil fertility and increased crop yields [25].

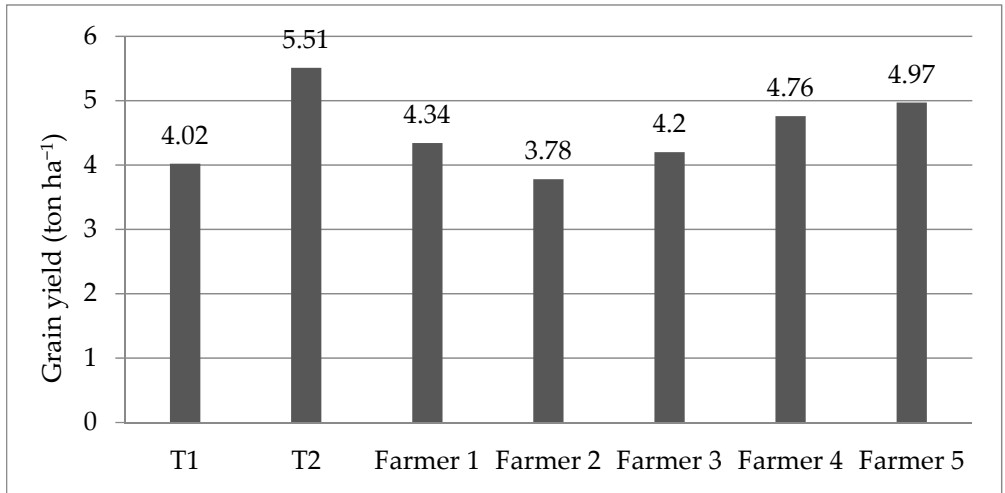

**Figure 1.** Comparison of research grain yields (ton ha$^{-1}$) and yields from local farmers.

The Azolla pinnata water fern has strong potential to be used as organic fertilizer. Azolla has a high nutrient content of N. A similar study in Indonesia on corn plants, with an application of Azolla at 5 tons ha$^{-1}$, revealed that the weight of the cobs could be increased by 20% [26]. Similarly, in a rice paddy study, adding Azolla at a rate of 10 tons ha$^{-1}$ increased the grain yield by 9.06% [17].

Usually, Azolla is immersed directly into the ground. In coastal areas that often experience inundation due to tidal flooding, this is less effective as the nutrients from Azolla are often washed away. Moreover, the salt content of the tidal flood increases absorption of nutrients in the soil, which in turn causes root system disturbance [27]. Therefore, to rationalize the process of nutrient absorption by plants, Azolla is extracted into a liquid biofertilizer, which is applied as foliar feeding. Fertilizer applied directly to leaves is more easily absorbed by plants, leading to quicker growth and greater production [8]. With the addition of other biological agents to the Azolla extract, the nutrients of the liquid biofertilizer can be further enriched.

In this study, the Inpari 43 and Mendawak varieties had a higher average grain yield compared to Mawar and Inpari 30, but they were not significantly different from the Inpara 03 variety. Farmer researchers confirmed that the varieties Inpari 43 and Mendawak were most susceptible to stress. With a yield potential of up to 6.96 ton ha$^{-1}$, Inpari 43 is well adapted to coastal locations with an altitude of 0–600 m ASL (meters above sea level) [28]. The Mendawak variety can appropriately adapt to inundation and saline conditions. Under inundation conditions of 0−10.8 cm and a salinity level of 1.49−7.36 ds m$^{-1}$, the Mendawak variety was still able to produce grain yields reaching 4.32 tons ha$^{-1}$, a higher yield than Inpara 02, Pelalawan, and Inpari 34 varieties [3]. Therefore, based on this research and previous supporting literature, the use of enriched Azolla extract (EAE) combined with

stress-tolerant varieties, namely Inpari 43 and Mendawak, is a viable farming practice to adapt to the negative consequences of climate change in coastal areas, such as inundation and salinity stresses.

## 4. Conclusions

This farmer-led research tested the use of enriched Azolla extract (EAE) at 10 L ha$^{-1}$ compared to the organic control variant. The replicated split-plot experiment showed that EAE significantly increased rice grain production by 37.06%. In addition, the average grain yield of five stress-tolerant rice varieties tested under EAE treatment (5.51 tons ha$^{-1}$) was greater than the grain yield in surrounding fields under the synthetic or mixed soil fertility strategies of local farmers (3.78–4.97 ton ha$^{-1}$). The highest grain yield was obtained by the variety Inpari 43 (5.90 tons ha$^{-1}$), followed by Mendawak (4.90 tons ha$^{-1}$), which showed no significant differences. This study concludes that EAE and selected rice varieties (Inpar 43 or Mendawak) are an appropriate agroecological technology able to increase the rice yield and the resilience of rice farming in saline and flood-prone coastal areas.

**Author Contributions:** Conceptualization, T.S., M.R.S., S.S. and K.A.; methodology, T.S., M.K.P., M.R.S. and S.S.; software, M.K.P.; validation, T.S.; formal analysis, M.K.P.; investigation, M.K.P. and K.A.; resources, T.S., M.R.S. and K.A.; data curation, M.K.P., writing—original draft preparation, T.S. and M.K.P.; writing—review and editing, T.S., M.R.S. and S.S.; visualization, M.K.P.; supervision, T.S, M.R.S. and S.S.; project administration, M.R.S. and K.A.; funding acquisition, T.S. and K.A. All authors have read and agreed to published version of the manuscript.

**Funding:** This research is part of the Climate Resilient Investigation and Innovation Project (CRAIIP) funded by the German Non-Governmental Organization Bread for the World (Second phase: 2019–2022).

**Institutional Review Board Statement:** Ethical review and approval were waived for this study, due to a prior agreement about the intellectual property rights and the study's operational procedures between involved farmer groups, academic institutions and the commissioning farmer organization JAMTANI.

**Informed Consent Statement:** Informed consent was obtained from all subjects involved in the study.

**Data Availability Statement:** The data and analyses from the current study are available from the corresponding authors upon reasonable request.

**Acknowledgments:** The authors acknowledge the Universitas Padjadjaran for supporting and providing laboratory facilities and financial support of the Academic Leadership Grant. The authors thank the farmer organization Indonesian Farmers Community Network (JAMTANI) for its contribution and collaboration during the experimental period. Our cordial thanks and sincere gratitude go to all research farmers who were actively involved in all stages of this participatory research.

**Conflicts of Interest:** The authors declare no conflict of interest.

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
