# Peer review of "Improving the Climate Resilience of Rice Farming in Flood-Prone Areas through Azolla Biofertilizer and Saline-Tolerant Varieties"

_sustainability, doi:10.3390/su132112308_

Round 1

Reviewer 1 Report

The subject of this manuscript will be an important study from a national point of view.  
However, I would like to point out the following several issues.
*Title is too long for the manuscript. You’re not much focused on climate change.    
Introduction; 
- Line 59~ : Why did you consider only one liquid fertilizer, Azolla Pinnata?
M&M;
- Line 82 : Is there any difference between EAE and AE?  
- Line 82~87 : It would be better to organize it in a table.
Results and discussion;  
- Line 108 : Need to give more information such as what kind of varieties were compared and so on. Is there a reason why table 1 is in the first place in results?   
- Table 2~6 : Does the ‘average of varieties’ value have any meaning? It would be better to make Table 2~6 into one or two tables.
- Line 202 : What is Y-axis title and unit? How much different between T2 and Farmer5?
- In general, there is a lack of reasonable objectivity for T1 and T2 treatments because it is expected that T2 with EAE would be effective.

Author Response

Thank you for the review that has been given. We are trying to improve so that this script becomes better based on the reviews you provide. 

Reviewer 2 Report

The paper deals with Enriched Azolla Extract As Liquid Biofertilizer And Selected Saline Tolerant Of Rice Variety To Increase The Resilient Of Rice Farming On Flooding Prone Coastal Area As Strategy In Adapting To Climate Change. 

English must be moderate revised.

Do not Capitalise all words of the title. Please revise also the title, something is not correct. Maybe: Enriched Azolla extract as a liquid biofertiliser and selected saline, tolerant to rice variety, to increase resistance to rice cultivation in coastal areas prone to floods, as a strategy to adapt to climate change. But it is much too long. Or: Enriched Azolla extract as a liquid biofertiliser and selected saline to increase rice resistance in coastal crops prone to floods as a strategy to adapt to climate change.

Keywords must to reflect the main characteristic words of the paper (usually reflected also by the title. So, I suggest the following keywords: enriched Azolla extract; liquid biofertiliser; selected saline; rice crops; flood; coastal area; climate changes; fertiliser. 

Introduction

I suggest to be added a paragraph discussing the relationship between fertilisers and climate changes, in order to facilitate a better understanding of it. You may check and refer to Bungau et al. Expatiating the impact of anthropogenic aspects and climatic factors on long term soil monitoring and management. Environ Sci. Pollut. Res. 2021, 202, 30528-30550. https://doi.org/10.1007/s11356-021-14127-7

L76-78. Please make this aim of the study more relevant. What makes special this study? Which is its novelty character or its special aspects? Why have the authors chosen this topic? What differentiate this paper from others in the same topic?   Table 1. legend: NS not ns (as in the 2nd column is denoted as NS). teratments must be corrected as Treatments. Tables 2, 3, 5, and 6. Please correct Treantments as Treatments. Tables 2-6. Column 4 - a, b, c, must be explained under the table; same for A and B, in each table. As the Instructions for authors of Sustainability journal request, please check and apply the rules for Acronyms/Abbreviations/Initialisms should be defined the first time they appear in each of three sections: the abstract; the main text; the first figure or table. When defined for the first time, the acronym/abbreviation/initialism should be added in parentheses after the written-out form. Table 7. Explain also all abbreviations. KCL or KCl?   The authors have chosen to have a single section of Results and Discussion, and I understood why. The part of Results is acceptable, but the Discussion part is very poor. It must be improved. You may find some useful data in Samuel et al. Effects of long term application of organic and mineral fertilizers on soil enzymes. Rev. Chim., 2018, 69(10),  2608-2612. https://doi.org/10.37358/RC.18.10.6590 ; Samuel et al. Enzymatic indicators of soil quality. J. Environ. Prot. Ecol. 2017, 18(3), 871-878; Samuel A.D., et al. Enzymological and physicochemical evaluation of the effects of soil management practices, Rev. Chim., 201768(10), 2243-2247. https://doi.org/10.37358/RC.17.10.5864  and in the paper I mentioned for Introduction. The Discussion part needs also to be better referenced and the References part must be improved.

Author Response

Thank you for the review that has been given. We are trying to improve so that this script becomes better based on the reviews you provide. I hereby attach a response to your review

Round 2

Reviewer 1 Report

From 'Response to Reviewer 1 Comment'

In No. 7, 'Line 202 : What is Y-axis title and unit? How much different between T2 and Farmer5?'

The author make a change(X-, Y-axis title) in response to the reviewer, but I could not find differences in the revision. Check it please.

I would expected that the author make error bars on the graphs for the question of 'How much different between T2 and Farmer5?'

Author Response

Thank you for the review, here's how to send a response to the review you gave

Reviewer 2 Report

The authors responded to all my requests.

Author Response

Thank you for the review you gave. This review helped me in improving this article.
